

# Mobile impurities in integrable models

**Andrew S. Campbell and Dimitri M. Gangardt⋆**

School of Physics and Astronomy, University of Birmingham, Edgbaston,
Birmingham, B15 2TT, United Kingdom

⋆ d.m.gangardt@bham.ac.uk

## Abstract

We use a mobile impurity or depleton model to study elementary excitations in one-dimensional integrable systems. For Lieb-Liniger and bosonic Yang-Gaudin models we express two phenomenological parameters characterising renormalised interactions of mobile impurities with superfluid background: the number of depleted particles, $N$ and the superfluid phase drop $\pi J$ in terms of the corresponding Bethe Ansatz solution and demonstrate, in the leading order, the absence of two-phonon scattering resulting in vanishing rates of inelastic processes such as viscosity experienced by the mobile impurities.



# 1  Introduction

Currently there is an ongoing interest in physics beyond Luttinger Liquid in one-dimensional systems [1]. An example of such behaviour is provided by systems featuring mobile impurities, representing one or few particles moving in a correlated background liquid. This situation was recently realised in ultra-cold atoms by either using atoms in different hyperfine states [2–4] or by creating a highly imbalanced mixture of different atoms [5,6]. By exploiting unprecedented control over parameters of the underlying Hamiltonian these experiments have uncovered rich dynamics of mobile impurities and have revived interest in their theoretical studies either in the context of liquid helium [7–9] or, more recently, in the context of one-dimensional quantum liquids [10, 11].

There are two main types of questions one can ask about these systems. The first one concerns dissipative dynamics of mobile impurities resulting from the strong coupling to environment [12–20], while the second type of questions deals with correlation properties of quantum liquids affected by the presence of mobile impurities [21–23]. These questions can be extended to one-component quantum liquids as their high-energy nonlinear excitations, like solitons, can be described as effective mobile impurities [24, 25].

Theoretically, the physics of mobile impurities is modelled by a single localised degree of freedom coupled to an extended environment. For sufficiently low temperatures and weak external fields the latter is only slightly perturbed so it can conveniently be represented by a collection of phonons in the form of Luttinger Liquid (LL) [26–30]. The coupling between the localised impurity and LL can be characterised by only two parameters: the number of background particles, $N$, expelled from the vicinity of impurity and the phase drop, $\pi J$, of the local superfluid order parameter. The existence of these parameters, called collective charges below, is attributed to the existence of two local conservation laws: the conservation of number of particles and conservation of momentum. The length scale separation between localised impurities and extended phonons allows for extraction of the values of collective charges from the response of the equilibrium dispersion relation $\varepsilon(k)$ of impurity-like excitations to small changes in background density and velocity fields and leads to a phenomenological depleton model [14, 16] for which $\varepsilon(k)$ is the only physical input.

This model was successful in predicting the rate of momentum exchange between the moving impurity and the phononic bath giving rise to viscous friction experiences by impurities [10, 11, 13]. A slightly different, but equivalent approach was used for determining relaxation rates toward thermal equilibrium in one dimensional systems due to mobile impurities [17, 31].

On the other hand, the contribution of mobile impurities to correlation functions leads to power law singularities near the excitation threshold $\varepsilon(k)$ [1]. It can therefore be expected that the corresponding edge exponents are determined by the same collective charges characterising the impurity-background interactions in the depleton model. The relation can indeed be established using the method of Kamenev and Glazman [23] based on Ref. [9] for extract-

ing the edge exponents from the magnon dispersion of a general Galilei invariant spin liquid. The method was further developed in Ref. [32] for a variety of Galilei invariant bosonic or fermionic models. A similar calculation based on the equilibrium dispersion $\varepsilon(k)$, as we show below, leads to the collective charges of Ref. [14], thus establishing their connection to the correlation properties of one-dimensional quantum liquids with mobile impurities.

For a general interacting system, the dispersion relation $\varepsilon(k)$ which determines both dynamics and correlations in systems with mobile impurities can be either calculated analytically using an appropriate perturbation scheme or obtained numerically. There exist, nonetheless, a class of exactly solvable one-dimensional systems, where the dispersion of excitations is known exactly in terms of Bethe Ansatz solution [33]. Important examples of such systems include Lieb-Liniger model [34] of bosons interacting via a delta function potential and the closely related bosonic Yang-Gaudin model [35, 36] of interacting spin one-half bosons.

The goal of this paper is to treat excitations of Lieb-Liniger and bosonic Yang-Gaudin models as mobile impurities and find their collective charges in terms of the corresponding Bethe Ansatz solutions. We were able to express collective charges directly in terms of shift functions (otherwise known as dressed phases in BA literature) of elementary excitations. This result reproduces the conjectured identity of chiral linear combinations of the collective charges, so called chiral phase shifts, with BA shift functions [22, 32, 37, 38]. Recently, the calculation of correlation functions in Lieb-Liniger model by the form-factor approach by Kitanine *et al.* [39] confirmed the expression of the edge exponents in terms of BA shift functions obtained in [38].

The expression of the collective charges in terms of BA shift functions allows us to find a very useful and transparent representation for $N$ and $J$ in terms of derivatives of the excitation's momentum with respect to BA parameters, see Eq. (32) and (41) below. These identities lead straightforwardly to exact vanishing of the phononic back scattering amplitude in the leading, two-phonon, order. This result should not come as a surprise given the exact integrability of the models, but it is far from obvious in the phenomenological approach based on mobile impurities. Up to now it could only be verified in a number of particular cases of weak and strong interactions [13, 25].

The paper is organised as follows. In Section 2 we review the depleton model containing collective charges $N$ and $J$ and give their expression in terms of the dispersion $\varepsilon(k)$. We demonstrate the relation between the collective charges and chiral phase shifts and use it to reinterpret various edge exponents in terms of $N$ and $J$. In Section 3 we review BA solution for elementary excitations of Lieb-Liniger and bosonic Yang-Gaudin models. Then we obtain exact relations between the collective charges $N$ and $J$ and solutions of BA equations for Lieb-Liniger model in Section 4 and for bosonic Yang-Gaudin model in Section 4.2. Finally, we discuss in Section 5 phononic back scattering amplitude and show that it vanishes identically for these models. We present conclusions and open questions in Section 6. Technical details needed for the proofs are delegated into four Appendices for clarity.

## 2 Depleton model, its collective charges and edge exponents of correlation functions near excitation thresholds

Depleton model is designed to describe mobile impurities within path intgral formalism and is defined by the action $\mathscr{S} = \mathscr{S}_{\text{ph}} + \mathscr{S}_{\text{d}}$, where

$$\mathscr{S}_{\text{ph}} = -\frac{1}{\pi} \int dt dx \left( \partial_x \vartheta \partial_t \varphi + \frac{v_s K}{2} (\partial_x \varphi)^2 + \frac{v_s}{2K} (\partial_x \vartheta)^2 \right), \qquad (1)$$

describes extended environment by using the slow hydrodynamic fields $\varphi(x, t)$ and $\vartheta(x, t)$. The latter is related to density as $n(x, t) = n + \rho(x, t) = n + \partial_x \vartheta / \pi$, where $n$ is the equilibrium

density. Correspondingly, the field $\varphi(x,t)$ describes the superfluid phase and is canonically conjugated to $\rho(x,t)$. The Luttinger Liquid action, Eq. (1), describes phonons with linear dispersion relation $\omega = v_s|k|$, where $v_s$ is speed of sound. The relative magnitude of fluctuations of the fields $\varphi$ and $\vartheta$ are controlled by the Luttinger parameter $K = \pi n/m v_s$, where $n$ is the average density and $m$ is the mass of the background atoms.

To describe a localised object with coordinate $X(t)$ and the canonically conjugate momentum $P(t)$ we consider action

$$\mathscr{S}_d = \int dt \left( P\dot{X} - H(P,N,J) - \dot{N}\varphi(X,t) - \dot{J}\theta(X,t) \right). \tag{2}$$

This action describes a sharp "kink" in the smooth configuration of the hydrodynamic fields located at $X(t)$ and parametrised by collective charges $N(t)$ and $J(t)$. The depleton charge $N(t)$ represents the number of background particles expelled from the vicinity of the depleton, therefore $\pi N(t)$ is the magnitude of the "kink" in the otherwise smooth configuration of $\vartheta$. Similarly, $\pi J(t)$ is the size of the "kink" in $\varphi$. In the absence of phonons, the charges take their equilibrium values $N(t) = N$, $J(t) = J$ obtained from the condition $\partial H/\partial N = \partial H/\partial J = 0$. The dispersion relation is the equilibrium value of the energy $H(P,N,J) = \varepsilon(P)$. It was shown in [14] that the equilibrium collective charges are given directly in terms of the dispersion[1],

$$N = \frac{1}{v^2(k) - v_s^2} \left( \frac{v_s K}{\pi} \frac{\partial \varepsilon(k)}{\partial n} + \frac{k}{m} v(k) \right), \tag{3}$$

$$J = \frac{1}{v^2(k) - v_s^2} \left( \frac{v(k)}{\pi} \frac{\partial \varepsilon(k)}{\partial n} + \frac{v_s}{K} \frac{k}{m} \right). \tag{4}$$

Here $v(k) = \partial \varepsilon(k)/\partial k$ is the velocity of the excitation.

In addition to describing dissipative dynamics of mobile impurities [14, 16] the depleton action $\mathscr{S}$ can be used to calculate leading power law behaviour of various correlation functions the vicinity of the excitation threshold $\omega \sim \varepsilon(k)$ as shown in Appendix A. In this case the path integral is dominated by the stationary value of the action $\mathscr{S}$ which becomes logarithmically large and leads to the power-law behaviour $\sim |\omega - \varepsilon(k)|^{-\mu}$, where the edge exponent

$$\mu = 1 - 2K \left[ \left( \frac{N}{2K} \right)^2 + \left( \frac{J}{2} \right)^2 \right] \tag{5}$$

is determined by the equilibrium values of the collective charges (3), (4).

The collective charges $N$ and $J$ can be connected with the parameters $\delta_\pm$, called chiral phase shifts, of the unitary transformation in the standard calculation [1] of the correlation functions using the X-ray edge approach [40–42]. In the latter, the chiral phase shifts characterise the discontinuities in the chiral phononic fields $\chi_\pm = \vartheta/\sqrt{K} \pm \sqrt{K}\varphi$ created by boundary condition changing operators [43, 44]. The charges $N, J$, on the other hand, characterise the discontinuities in the fields $\vartheta$ and $\varphi$. The simplest way to establish connection between these two sets of parameters is by comparing Eqs. (3), (4) with those of the phase shifts $\delta_\pm$

$$\frac{\delta_\pm}{\pi} \pm \frac{1}{\sqrt{K}} = \frac{1}{v(k) \mp v_s} \left( \frac{\sqrt{K}}{\pi} \frac{\partial \varepsilon(k)}{\partial n} \pm \frac{1}{\sqrt{K}} \frac{k}{m} \right), \tag{6}$$

first obtained in Ref. [23, 32] in terms of the mobile impurity dispersion relation. It is easy to see that the phase shifts and the collective charges are related by

$$\frac{\delta_\pm(k)}{\pi} \pm \frac{1}{\sqrt{K}} = \sqrt{K}J \pm \frac{N}{\sqrt{K}}. \tag{7}$$

---

[1]Ref. [14] uses notations $N, \Phi$, where $N$ has the same meaning as in this paper and $\Phi = \pi J$.

The expression (5) reproduces known results for edge exponents if rewrite it in terms of the chiral phase shifts as

$$
\begin{aligned}
\mu &= 1 - \frac{1}{2}\left(\frac{1}{\sqrt{K}} + \frac{\delta_+ - \delta_-}{2\pi}\right)^2 - \frac{1}{2}\left(\frac{\delta_+ + \delta_-}{2\pi}\right)^2 \\
&= 1 - \left(\frac{\delta_+}{2\pi} + \frac{1}{2\sqrt{K}}\right)^2 - \left(\frac{\delta_-}{2\pi} - \frac{1}{2\sqrt{K}}\right)^2 .
\end{aligned}
\tag{8}
$$

The first line reproduces exactly the results for the edge exponents $\mu_{1,2}$ of Dynamic Structure Factor (DSF) of interacting bosons, Eq. (16) of Ref. [38], while the second line is Eq. (13) of Ref. [23] for the exponent $\mu_m$ of Dynamic Spin Structure Factor (DSSF) of a general spin one half quantum liquid. To reproduce edge exponents $\overline{\mu_\pm}$ of the spectral function $A(k, \omega)$ in Ref. [38] one notices that in this case, in addition to creating a depleton, one removes exactly one particle from the system, so one have to replace the number of missing particles $N$ by $N + 1$. In terms of the phase shifts $\delta_\pm$ this leads to

$$
\overline{\mu_\pm} = 1 - \frac{1}{2}\left(\frac{\delta_+ - \delta_-}{2\pi}\right)^2 - \frac{1}{2}\left(\frac{\delta_+ + \delta_-}{2\pi}\right)^2 ,
\tag{9}
$$

which reproduces exactly Eq. (17) of [38].

Similarly, to calculate $\underline{\mu}_\pm$ one replaces $N$ by $N-1$, which corresponds to adding one particle so that

$$
\underline{\mu}_\pm = 1 - \frac{1}{2}\left(\frac{2}{\sqrt{K}} + \frac{\delta_+ - \delta_-}{2\pi}\right)^2 - \frac{1}{2}\left(\frac{\delta_+ + \delta_-}{2\pi}\right)^2 ,
\tag{10}
$$

again in full accordance with Eq. (18) of Ref. [38].

The same logic applies for calculating the edge exponents away from the fundamental zone $0 < k < 2\pi n$. Consider the lower edge, the Lieb II mode $\varepsilon(k) = \varepsilon_2(k)$ as an example. A remarkable fact with far fetching physical consequences is that $\varepsilon_2(q)$ is a periodic function, $\varepsilon_2(q + 2\pi n) = \varepsilon_2(q)$. This is due to the fact that the momentum difference $2\pi n$ results from the presence of additional uniform background supercurrent and the corresponding phase winding of $2\pi$. The energies of the state with the supercurrent and without it are the same in the thermodynamic limit. The additional phase winding $2\pi l$ ($l$ is an integer) affects the value of $J$ and can be taken into account by the substitution $J \to J - 2l$ in Eq (5) so that the resulting edge exponent becomes

$$
\begin{aligned}
\mu_l &= 1 - 2K\left[\left(\frac{N(k)}{2K}\right)^2 + \left(\frac{J(k)}{2} - l\right)^2\right] \\
&= 1 - \frac{1}{2}\left(\frac{1}{\sqrt{K}} + \frac{\delta_+ - \delta_-}{2\pi}\right)^2 - \frac{1}{2}\left(\frac{\delta_+ + \delta_-}{2\pi} + 2l\sqrt{K}\right)^2 ,
\end{aligned}
\tag{11}
$$

which is the bosonic version of Eq. (87) of Ref. [1].

Thus the expression (5) obtained within the depleton formalism nicely combines previous results on edge exponents of dynamical correlation functions obtained using the mobile impurity model. The collective charges $N$ and $J$ entering this expression can be obtained from phenomenological dispersion of the mobile impurity excitations by Eqs. (3), (4). Below we obtain the collective charges for integrable models using excitation dispersion from their Bethe Ansatz solutions.

# 3   Bethe Ansatz solution and elementary excitations of Lieb-Liniger and bosonic Yang-Gaudin models

Up to now our discussion of the effective mobile impurity with dispersion relation $\varepsilon(k)$ was purely phenomenological. We now turn to the models where the dispersion relation and, consequently, the parameters $N, J$ determining the edge exponents can be obtained analytically by Bethe Ansatz. We consider Lieb-Liniger model [34] describing one-dimensional bosons interacting with a short-range potential $V(x) = c\delta(x)$ which is arguably the most studied continuous model solved by Bethe Ansatz. Generalising this model to two species of bosonic particles of the same mass and equal interaction couplings leads to bosonic Yang-Gaudin model [35,36]. This model can be conveniently formulated using effective spin $1/2$ particles. Its ground state is fully polarised [45] and is identical to that of Lieb-Liniger model. Below we present main equations which allow us to obtain dispersion relation $\varepsilon(k)$ used for calculation of the collective charges. All results of this Section can be found in Refs. [33,46] and are reproduced here to make presentation self-contained.

The ground state of Lieb-Liniger model is characterised by the density of quasi-momenta $\rho(\nu)$ found from the equation

$$\rho(\nu) - \frac{1}{2\pi}\int_{-q}^{q} K(\nu-\mu)\rho(\mu)\,\mathrm{d}\mu = \frac{1}{2\pi}. \tag{12}$$

Here $K(\lambda) = \partial\theta/\partial\lambda = 2c/(c^2 + \lambda^2)$, where $\theta(\lambda) = 2\arctan(\lambda/c)$ is the scattering phase shift. We use units in which the mass of the particles is $m = 1/2$ and $\hbar = 1$. In these units the coupling constant $c$ and quasi-momenta $\lambda$ have dimensions of velocity.

The "Fermi momentum" $q$ limits the support of the ground state density and is determined from the normalisation condition

$$\int_{-q}^{q} \rho(\nu)\mathrm{d}\nu = n. \tag{13}$$

The excitations of Lieb-Liniger model are in one to one correspondence with those of free fermions and consist of either a particle excited above the Fermi sea (Lieb I), so its quasi-momentum $\lambda$ is constrained by $|\lambda| > q$ or a hole (Lieb II) with $-q < \lambda < q$ [47]. The difference from free fermions is that the positions of the ground state quasi-momenta are shifted as a result of the excitation and these shifts contribute collectively to momentum and energy. This effect is characterised by the *shift function* $F(\nu|\lambda)$ which obeys the equation

$$F(\nu|\lambda) - \frac{1}{2\pi}\int_{-q}^{q} K(\nu-\mu)F(\mu|\lambda)\,\mathrm{d}\mu = \frac{\theta(\nu-\lambda)}{2\pi}. \tag{14}$$

Taking the limit $\lambda \to \pm\infty$ and comparing with Eq. (12) it follows that $F(\nu|\pm\infty) = \mp\pi\rho(\nu)$. The shift function represents the relative change of the ground state quasi-momenta $\delta\nu\rho(\nu) = \mp(F(\nu|\lambda) + \pi\rho(\nu))$ as a result of a particle/hole with quasi-momentum $\lambda$. Here and below the upper sign corresponds to a particle-like (Lieb I) excitation while the lower sign corresponds to a hole-like (Lieb II) excitation. Some useful properties of the shift function $F(\lambda|\nu)$ are summarised in Appendix B.

The momentum and energy of Lieb I, II excitations are given by

$$\pm k(\lambda) = \lambda - \pi n - \int_{-q}^{q} F(\nu|\lambda)\,\mathrm{d}\nu, \tag{15}$$

$$\pm\varepsilon(\lambda) = \lambda^2 - h - 2\int_{-q}^{q} \mu F(\mu|\lambda)\,\mathrm{d}\mu. \tag{16}$$

The integrals in the right hand side represent the collective contribution of the displaced momenta in the "Fermi sea". Eliminating $\lambda$ from Eqs. (15) and Eq. (16) leads to the dispersion relation $\varepsilon(k)$. Apart from momentum, the dispersion relation $\varepsilon(k)$ depends on the density $n$ via the limiting momentum $q$ fixed by normalisation (13). The chemical potential $h$ is then fixed from the condition $\varepsilon(q) = \varepsilon(-q) = 0$. As both signs in Eq. (15) should provide the same result for $\lambda = q$, it is clear that $k(q) = 0$. Since for a hole the momentum increases with decreasing $\lambda$ one can show that $k(0) = \pi n$ and $k(-q) = 2\pi n$. For a particle $k(-q) = -2\pi n$. There is an alternative way to express the momentum,

$$\pm k(\lambda) = \lambda - \pi n + \int_{-q}^{q} \theta(\lambda - \nu)\rho(\nu)\,\mathrm{d}\nu. \tag{17}$$

Similarly, the energy of excitations $\epsilon(\lambda) = \pm\varepsilon(\lambda)$ for particles ($|\lambda| > q$) / holes ($|\lambda| < q$) can be found from the equation

$$\epsilon(\lambda) - \frac{1}{2\pi}\int_{-q}^{q} K(\lambda - \mu)\epsilon(\mu)\,\mathrm{d}\mu = \lambda^2 - h. \tag{18}$$

The equivalence of (17), (18) and (15), (16) was first demonstrated in Ref. [33]. We reproduce it for convenience in Appendix D.1.

In the bosonic Yang-Gaudin model in addition to Lieb I,II excitations there is another type of excitations corresponding to a spin flip of one of the particles. The flipped spin particle is introduced with quasi-momentum $\lambda$ causing the change in the quasi-momenta $\delta\nu\rho(\nu) = \tilde{F}(\nu|\lambda) + \pi\rho(\nu)$. The shift function $\tilde{F}(\nu|\lambda)$ is found from the equation

$$\tilde{F}(\nu|\lambda) - \frac{1}{2\pi}\int_{-q}^{q} K(\nu - \mu)\tilde{F}(\mu|\lambda)\,\mathrm{d}\mu = \frac{\theta(2\nu - 2\lambda)}{2\pi}. \tag{19}$$

This equation differs from Eq. (14) by the factor two in the argument of the bare phase shift. The momentum of the magnon excitation is obtained by the shift of the ground-state quasi-momenta Eqs. (15), (17)

$$\tilde{k}(\lambda) = \pi n + \int_{-q}^{q} \tilde{F}(\nu|\lambda)\,\mathrm{d}\nu = \pi n + \int_{-q}^{q} \rho(\nu)\theta(2\nu - 2\lambda)\,\mathrm{d}\nu. \tag{20}$$

The last equality follows from Eq. (19). The corresponding energy of a magnon can be similarly expressed as

$$\tilde{\varepsilon}(\lambda) = 2\int_{-q}^{q} \nu\tilde{F}(\nu|\lambda)\,\mathrm{d}\nu = \frac{1}{2\pi}\int_{-q}^{q} \partial_\nu\epsilon(\nu)\theta(2\nu - 2\lambda)\,\mathrm{d}\nu = -\frac{1}{\pi}\int_{-q}^{q} \epsilon(\nu)K(2\nu - 2\lambda)\,\mathrm{d}\nu, \tag{21}$$

where $\epsilon$ is the solution of Eq. (18) and the last equality is obtained by integrating by parts.

# 4 Collective charges and shift functions in integrable models

The parametric expressions (15), (16) and (20), (21) of the previous section allows us to obtain the dispersion relations of excitation $\varepsilon(k)$ in Lieb-Liniger and $\tilde{\varepsilon}(\tilde{k})$ in bosonic Yang-Gaudin models in terms of shift functions $F(\nu|\lambda)$ and $\tilde{F}(\nu|\lambda)$ correspondingly. It is therefore expected that the same Bethe ansatz shift function determine the collective charges $N, J$ or their chiral combinations $\delta_\pm(k)$. Indeed, the previous studies [22, 38] suggested the following particularly simple and physically appealing relation

$$\frac{\delta_\pm(k)}{2\pi} = F(\pm q|\lambda(k)) + \pi\rho(\pm q) \tag{22}$$

for excitations of Lieb-Liniger model and a similar expression for magnon excitation of bosonic Yang-Gaudin model. The conjecture (22) is based either on identification of $\delta_\pm(k)$ with the parameters of a boundary changing operator of background particles [38] or on comparing the finite size spectra [22, 37, 48, 49]. To the best of our knowledge, the direct proof of Eqs. (22) for integrable models is still lacking and the consistency of the thermodynamic definition Eq. (6) with the relation based on BA shift functions was established only numerically. It should be noted, however, that the relation (22), being substituted into Eq. (5) leads to the edge exponent identical[2] to that obtained within the form-factor method in Ref. [39].

Below we provide the direct proof of Eq. (22). In the Subsection 4.1 we deal with Lieb I and II excitations of Lieb-Liniger model. The relation similar to Eq. (22) for bosonic Yang-Gaudin model is stated and demonstrated in Subsection 4.2.

## 4.1 Lieb I and II excitations of Lieb-Liniger model

Our proof is based on the following expressions for the the derivatives of the dispersion relation with respect to the "natural" parameters $\lambda$ and $q$. Defining for simplicity $F_\pm = F(\pm q | \lambda)$ we show in Appendices C,D that

$$\partial_q \epsilon(\lambda) = -v_s \left( 1 + \sqrt{K} \left( F_+ - F_- \right) \right), \tag{23}$$

and

$$\partial_\lambda \epsilon(\lambda) = \pm 2k(\lambda) + 2\pi n \left( 1 + \frac{1}{\sqrt{K}} \left( F_+ + F_- \right) \right). \tag{24}$$

We also show there that

$$\pm \partial_q k = -K \left( 1 + \frac{1}{\sqrt{K}} \left( F_+ + F_- \right) \right) \tag{25}$$

and by differentiating both sides of Eq. (17) and comparing the result with Eq. (12) we have

$$\pm \partial_\lambda k(\lambda) = 2\pi \rho(\lambda) = 1 + F(\lambda | q) - F(\lambda | -q), \tag{26}$$

where we have used Eq. (58). We will also need the relation

$$\partial_q n = \frac{K}{\pi}. \tag{27}$$

proven in Appendix D.1.

We transform the derivatives (23), (24) of the dispersion relation with respect to the natural variables $\lambda$ and $q$ to those with respect to density $n$ and momentum $k$. This can be achieved using Eqs. (25), (26) and (27). Re-introducing the particle's mass $m$ and going back to $\varepsilon = \pm \epsilon$ for particles/holes we rewrite Eqs. (23), (24) as

$$\partial_\lambda \varepsilon = \frac{k}{m} \pm v_s \sqrt{K} \left( \sqrt{K} + F_+ + F_- \right) = \partial_\lambda k \, \partial_k \varepsilon = \pm v(k) \left( 1 + \sqrt{K} \left( F_+ - F_- \right) \right),$$

$$\partial_q \varepsilon = \mp v_s \left( 1 + \sqrt{K} \left( F_+ - F_- \right) \right) = \partial_q n \, \partial_n \varepsilon + \partial_q k \, \partial_k \varepsilon = \frac{K}{\pi} \partial_n \varepsilon \mp v(k) \sqrt{K} \left( \sqrt{K} + F_+ + F_- \right).$$

If one identifies in these equations, accordingly to Eq. (22), the combinations

$$\frac{\delta_+ + \delta_-}{2\pi} = 2\pi \rho(q) + F_+ + F_- = \sqrt{K} + F_+ + F_-,$$

$$\frac{\delta_+ - \delta_-}{2\pi} = F_+ - F_-,$$

---

[2]There is an overall minus sign due to a different definition of the edge exponent. Note also the difference in our definition of BA shift functions and the one used in Ref. [39].

and solves for $\delta_\pm$ using the upper sign for particles one obtains precisely Eqs. (6).

The collective charges $N, J$ can be obtained using Eq. (7) with the result

$$N = 1 + \sqrt{K}\big(F(q|\lambda) - F(-q|\lambda)\big) = 1 - F(\lambda|q) + F(\lambda|-q) = 2\pi\rho(\lambda)\,, \tag{28}$$

$$J = 1 + \frac{1}{\sqrt{K}}\big(F(q|\lambda) + F(-q|\lambda)\big) = 1 - F(\lambda|q) - F(\lambda|-q)\,. \tag{29}$$

For $\lambda = \pm q$ we have $N = 2\pi\rho(\pm q) = \sqrt{K} = 1/J$.

In the case of a hole-like Lieb II excitation, one has $k \to -k$, $\varepsilon \to -\varepsilon$, but $v(k) = \partial\varepsilon/\partial k$ is unchanged. The expressions (3), (4) become

$$N = \frac{1}{v_s^2 - v^2(k)}\left(\frac{v_s K}{\pi}\frac{\partial\varepsilon(k)}{\partial n} + v(k)\frac{k}{m}\right) \tag{30}$$

$$J = \frac{1}{v_s^2 - v^2(k)}\left(\frac{v(k)}{\pi}\frac{\partial\varepsilon(k)}{\partial n} + \frac{v_s}{K}\frac{k}{m}\right)\,. \tag{31}$$

Comparing Eqs. (28), (29) with Eqs. (25) and (26) allows one to rewrite the collective charges for both Lieb I (particles) and Lieb II (holes) via partial derivatives of momentum $k(\lambda; q)$ as

$$N = \pm\frac{\partial k}{\partial\lambda}\,, \qquad J = \mp\frac{1}{K}\frac{\partial k}{\partial q}\,. \tag{32}$$

We are not aware of any previous studies discovering these nontrivial relations.

## 4.2 Magnon excitation in bosonic Yang-Gaudin model

For bosonic Yang-Gaudin model the phenomenological approach was used in Ref. [23] to express the phase shifts of the magnon excitation in terms of the magnon momentum $\tilde{k}$ and derivatives of the magnon dispersion relation $\tilde{\varepsilon}(\tilde{k})$ leading to the result (6) in which one replaces $k \to \tilde{k}$, $\varepsilon \to \tilde{\varepsilon}$ and the group velocity $v(k) \to \tilde{v}(\tilde{k}) = \partial_{\tilde{k}}\tilde{\varepsilon}$. In later work [22] a relation between the phase shifts of the magnon excitation and Bethe Ansatz shift function, similar to Eq. (22) was proposed. In the notation of Ref. [23], and using a slightly different sign convention, this relation is given by

$$\frac{\delta_\pm(k)}{2\pi} \pm \frac{1}{2\sqrt{K}} = -\tilde{F}(\pm q|\lambda(k)) - \frac{\sqrt{K}}{2}\,. \tag{33}$$

To prove this relation by analogy to the case of the Lieb-Liniger model we need derivatives of the dispersion relation:

$$\partial_\lambda\tilde{\varepsilon} = \frac{\tilde{k}(\lambda)}{m} - v_s\sqrt{K}\big(\sqrt{K} + \tilde{F}(q|\lambda) + \tilde{F}(-q|\lambda)\big)\,, \tag{34}$$

$$\partial_q\tilde{\varepsilon} = v_s\sqrt{K}\big(\tilde{F}(q|\lambda) - \tilde{F}(-q|\lambda)\big)\,, \tag{35}$$

and derivatives of the momentum:

$$\partial_\lambda\tilde{k} = -\sqrt{K}\big(\tilde{F}(q|\lambda) - \tilde{F}(-q|\lambda)\big)\,, \tag{36}$$

$$\partial_q\tilde{k} = \sqrt{K}\big(\sqrt{K} + \tilde{F}(q|\lambda) + \tilde{F}(-q|\lambda)\big)\,, \tag{37}$$

proven in Appendix D.3. Substituting Eqs. (34), (35), (36) and (37) into the following chain rules:

$$\partial_\lambda\tilde{\varepsilon} = \partial_{\tilde{k}}\tilde{\varepsilon}\,\partial_\lambda\tilde{k}\,,$$
$$\partial_q\tilde{\varepsilon} = \partial_n\tilde{\varepsilon}\partial_q n + \partial_{\tilde{k}}\tilde{\varepsilon}\,\partial_q\tilde{k}\,,$$

and identifying the phase shifts via Eq. (33) gives

$$\frac{\delta_\pm(k)}{\pi} \pm \frac{1}{\sqrt{K}} = -\sqrt{K}J \mp \frac{N-1}{\sqrt{K}} = \frac{1}{\pm\tilde{v}(\tilde{k})-v_s}\left(\frac{1}{\sqrt{K}}\frac{\tilde{k}}{m} \pm \frac{\sqrt{K}}{\pi}\frac{\partial\tilde{\varepsilon}(\tilde{k})}{\partial n}\right), \qquad (38)$$

in full agreement with the results of Ref. [23]. The collective charges are

$$N = \frac{1}{v_s^2 - \tilde{v}^2(\tilde{k})}\left(\frac{\tilde{k}-m\tilde{v}(\tilde{k})}{m}\tilde{v}(\tilde{k}) + \frac{Kv_s}{\pi}\left(\frac{\partial\tilde{\varepsilon}}{\partial n} + \frac{\pi v_s}{K}\right)\right), \qquad (39)$$

$$J = \frac{1}{v_s^2 - \tilde{v}^2(\tilde{k})}\left(\frac{v_s}{K}\frac{\tilde{k}}{m} + \frac{\tilde{v}(\tilde{k})}{\pi}\frac{\partial\tilde{\varepsilon}}{\partial n}\right), \qquad (40)$$

as expected from Ref. [14]. Comparing Eqs. (30) and (39) we see that there are additional terms representing the bare momentum $m\tilde{v}(\tilde{k})$ of the impurity and its chemical potential $\pi v_s/K = mv_s^2/n = \partial\mu/\partial n$. These additional terms cancel each other in Eq. (40) .

In terms of the magnon shift functions, the collective charges are

$$N = 1 + \sqrt{K}\left(\tilde{F}(q|\lambda) - \tilde{F}(-q|\lambda)\right),$$

$$J = 1 + \frac{1}{\sqrt{K}}\left(\tilde{F}(q|\lambda) + \tilde{F}(-q|\lambda)\right).$$

Again, comparing these expressions with Eqs. (36) and (37) allows to rewrite the collective charges via partial derivatives of momentum as

$$N - 1 = -\frac{\partial\tilde{k}}{\partial\lambda}, \qquad J = \frac{1}{K}\frac{\partial\tilde{k}}{\partial q}. \qquad (41)$$

## 5  Phonon backscattering amplitude

It was shown in Ref. [14] that the viscous friction force acting on a moving impurity due to two-phonon processes is proportional to the squared absolute value of the phonon backscattering amplitude $\Gamma_{+-}$. It is expected that in integrable systems this amplitude vanishes due to the existence of infinitely many conservation laws. An example supporting this statement is provided by a dark soliton, which is a hole-like excitation of the Lieb-Liniger model in the weakly interacting regime and which was shown to have infinite life-time in Refs. [25, 50]. Another example of a non-decaying excitation is a spin-flipped particle (magnon) in bosonic Yang-Gaudin model, as was shown in Ref. [13] in the limit of weak and strong interactions. Below we prove that to the leading two-phonon order the backscattering amplitude vanishes identically for Lieb-Liniger and bosonic Yang-Gaudin models for *any value of interactions*.

The expression for the back scattering amplitude was obtained in Ref. [14] as the following combination of partial derivatives of the collective charges

$$\Gamma_{+-} = \frac{\pi}{v_s}\left(\left(N - \frac{M}{m}\right)\partial_k J - J\partial_k N - \frac{1}{\pi}\partial_n N\right). \qquad (42)$$

Here $M$ is the mass of the added impurity particle. As it stands, Eq. (42) is valid only for subsonic excitations. In our case they are the hole excitation of Lieb-Liniger model for which $M = 0$ since there is no additional particle and the magnon of bosonic Yang-Gaudin model for which $M = m$. For the supersonic particle-like excitation of Lieb-Liniger model the expression for the back scattering amplitude must be modified as explained below.

We start with a hole-like excitation for which $\Gamma_{+-}$ is given by

$$\Gamma_{+-} = \frac{\pi}{v_s}\left(N\partial_k J - J\partial_k N - \frac{1}{\pi}\partial_n N\right). \tag{43}$$

We convert the partial derivatives with respect to $k$ and $n$ into the partial derivatives with respect to the "natural" variables using

$$\partial_\lambda = \frac{\partial k}{\partial \lambda}\partial_k = \pm 2\pi\rho(\lambda)\partial_k = \pm N\partial_k \,,$$

$$\partial_q = \frac{\partial k}{\partial q}\partial_k + \frac{\partial n}{\partial q}\partial_n = \mp K\left(1 + \frac{1}{\sqrt{K}}\left(F(q|\lambda) + F(-q|\lambda)\right)\right)\partial_k + \frac{K}{\pi}\partial_n$$

$$= \mp KJ\partial_k + \frac{K}{\pi}\partial_n \,,$$

where we have used Eqs. (28) and (29). Inverting the above expressions we get

$$\partial_k = \pm\frac{1}{N}\partial_\lambda \,, \tag{44}$$

$$\partial_n = \frac{\pi J}{N}\partial_\lambda + \frac{\pi}{K}\partial_q \,, \tag{45}$$

which can be used (with the lower sign) in Eq. (43). It leads to a particularly simply looking result

$$-\left(\frac{v_s}{\pi}\right)\Gamma_{+-} = \partial_\lambda J + \frac{1}{K}\partial_q N = 0, \tag{46}$$

as a direct consequence of the relations (32).

For the particle-like excitations (Lieb I mode) the expression (43) must be modified by replacing $N \to -N$ and $J \to -J$ (*cf.* Eqs. (30), (31) and Eqs. (3), (4)). In combination with the upper sign in Eq. (44) this leads to

$$\left(\frac{v_s}{\pi}\right)\Gamma_{+-} = \partial_\lambda J + \frac{1}{K}\partial_q N = 0 \,.$$

Finally, for the magnon excitation of bosonic Yang-Gaudin model we can simply replace $N$ by $N-1$ and $k$ by $\tilde{k}$ in Eq. (43), (44), (45) and (46).

# 6 Summary and conclusions

To summarise, we have obtained the collective charges $N$ and $J$ of mobile impurities in integrable models directly in terms of the corresponding BA shift functions. Our method relies on the Bethe Ansatz solution which provides a unique parametrisation of the elementary excitations in these models by their rapidity as well as the parameter which limits the extent of the ground state rapidity distribution.

This parametrisation is expressed via Bethe Ansatz shift functions and allows for exact calculation of derivatives of the excitation energy with respect to momentum and density which enter the phenomenological expressions for the scattering phase shifts and collective charges found in earlier works.

As a byproduct we have found a novel expression for the collective charges of the effective mobile impurity model in terms of the partial derivatives of the impurity momentum with respect to the Bethe Ansatz parameters mentioned above.

A straightforward consequence of these relations is the absence of phonon scattering off mobile impurities in the leading two-phonon order for all values of interaction parameters.

This absence was previously conjectured based on approximate calculation in the limiting cases of weak and strong interactions. The proof of the expected absence of inelastic processes beyond two-phonon rate is an obvious extension of our work.

We expect that our methods can be generalised to study excitations in other models soluble by nested BA, such as, the fermionic Hubbard model and integrable spin chains. The lack of Galilean invariance in lattice models can be dealt with following techniques of Ref. [17].

## Acknowledgements

We are grateful to Adilet Imambekov for his encouragement to publish these results and dedicate this work to his memory.

## A Semiclassical calculation of power-law edge exponents using depleton model

We are interested in dynamical correlation functions of one-dimensional bosons. In particular we consider the zero-temperature Dynamic Structure Factor (DSF),

$$S(k,\omega) = \int dx dt \, e^{i\omega t - ikx} \langle \rho(x,t)\rho(0,0)\rangle, \tag{47}$$

and spectral function $A(k,\omega) = -\frac{1}{\pi} \text{Im} \, G(k,\omega) \, \text{sgn} \, \omega$, where

$$G(k,\omega) = -i \int dx dt \, e^{i\omega t - ikx} \left\langle \mathcal{T} \, \Psi(x,t)\Psi^\dagger(0,0) \right\rangle. \tag{48}$$

is the Green's function. Here $\Psi(x,t)$ and $\rho(x,t) = \Psi^\dagger(x,t)\Psi(x,t)$ are boson annihilation and density operators, and $\mathcal{T}$ denotes time ordering. In the case where bosons have two internal states $a, b$ we can define spin operators $s_+ = s_-^\dagger = \Psi_a^\dagger(x,t)\Psi_b(x,t)$ and the corresponding Dynamic Spin Structure factor (DSSF),

$$S_{\text{spin}}(k,\omega) = \int dx dt \, e^{i\omega t - ikx} \langle s_+(x,t)s_-(0,0)\rangle. \tag{49}$$

It is well known (see Ref. [1]) that for various one-dimensional models the dynamical correlation functions have a rich structure in the $(k,\omega)$ plane. In particular, they exhibit power law singularities in vicinity of dispersion curve $\varepsilon(k)$ of elementary excitations,

$$S(k,\omega), S_{\text{spin}}(k,\omega), A(k.\omega) \sim \left| \frac{1}{\omega - \varepsilon(k)} \right|^\mu. \tag{50}$$

This power law behaviour can be obtained by semiclassically evaluating the path integral

$$\langle \mathcal{O}(x_2,t_2)\mathcal{O}^\dagger(x_1,t_1)\rangle = \mathcal{Z}^{-1} \int \mathcal{D}\Psi \, e^{i\mathcal{S}} \, \mathcal{O}(x_2,t_2)\mathcal{O}^\dagger(x_1,t_1), \qquad \mathcal{Z} = \int \mathcal{D}\Psi \, e^{i\mathcal{S}}. \tag{51}$$

with the action $\mathcal{S} = \mathcal{S}_{\text{ph}} + \mathcal{S}_{\text{d}}$, see Eqs. (1), (2). As we shall see below the semiclassical method is justified by the smallness of the energy difference $\omega - \varepsilon(k)$ leading to a logarithmically large action.

Following Iordanskii and Pitaevskii [51] we divide the time countour into three intervals, $(-\infty, t_1), [t_1, t_2], (t_2, +\infty)$. The kinematic constraints dictate that in this vicinity of the

excitation energy $\varepsilon(k)$ it is enough to consider configurations consisting of only one depleton [14,16] propagating in the time interval $[t_1, t_2]$. Then the stationary configuration of the fields dominating the functional integral can now be described as follows: the depleton appears at $t_1$ at $x_1$, propagates as a point-like particle with a constant velocity $V = (x_2 - x_1)/(t_2 - t_1)$ and disappears at $t_2$ at $x_2$. Its trajectory is given by $X(t) = x_1 + Vt$. This is reflected in the following behaviour of the collective charges

$$\dot{N}(t) = N[\delta(t - t_1) - \delta(t - t_2)], \tag{52}$$

$$\dot{J}(t) = J[\delta(t - t_1) - \delta(t - t_2)]. \tag{53}$$

Here the values $N$ and $J$ correspond to their corresponding equilibrium values calculated from Eqs. (3), (4) in which the momentum $k$ is such that $v(k) = V$.

The corresponding stationary configuration of the phononic variables is determined by solving the wave equation with the source terms

$$\partial_t \partial_x \vartheta + v_s K \partial_x^2 \varphi = \pi \dot{N} \delta(x - X(t))$$

$$\partial_t \partial_x \varphi + (v_s/K) \partial_x^2 \vartheta = \pi \dot{J} \delta(x - X(t)).$$

Solving these equations leads to logarithmic behaviour of the action

$$\mathscr{S}_{\mathrm{ph}} + N[\varphi(x_2, t_2) - \varphi(x_1, t_1)] + J[\vartheta(x_2, t_2) - \vartheta(x_1, t_1)]$$
$$= \frac{\Lambda_+^2}{2\pi \mathrm{i}} \ln\left(\frac{v_s t_{21} - x_{21}}{\xi}\right) + \frac{\Lambda_-^2}{2\pi \mathrm{i}} \ln\left(\frac{v_s t_{21} + x_{21}}{\xi}\right),$$

where $t_{21} = t_2 - t_1$, $x_{21} = x_2 - x_1$ and we have introduced symmetric and antisymmetric combinations

$$\Lambda_\pm = \sqrt{\frac{\pi}{2}}\left(\sqrt{K}J \pm \frac{1}{\sqrt{K}}N\right). \tag{54}$$

of the depleton collective charges.

To proceed with the correlation function we need the Fourier transform of the type

$$\int \mathrm{d}x \mathrm{d}t \, \mathrm{e}^{-\mathrm{i}kx + \mathrm{i}\omega t + \mathrm{i}\mathscr{S}_{\mathrm{d}}(x,t)} \left(\frac{\xi}{v_s t - x + \mathrm{i}\eta}\right)^{\frac{\Lambda_+^2}{2\pi}} \left(\frac{\xi}{v_s t + x + \mathrm{i}\eta}\right)^{\frac{\Lambda_-^2}{2\pi}},$$

where an infinitesimal imaginary part was added to avoid power-law singularities. The action of depleton $\mathscr{S}_{\mathrm{d}}(x,t)$ is considered as function of time and coordinate, and its full differential obeys

$$\mathrm{d}\mathscr{S}_{\mathrm{d}}(x,t) = P\mathrm{d}x - H\mathrm{d}t. \tag{55}$$

The integral over coordinate $x$ is performed by stationary phase method, which locks the momentum $P$ of the depleton to the externally imposed value $P = \partial \mathscr{S}_{\mathrm{d}}/\partial x = k$ and energy $H = \varepsilon(k)$. The velocity of the depleton is now a function of momentum and its stationary trajectory is $x = v(k)t$. The collective charges $N$ and $J$ and their combinations $\Lambda_\pm$ are functions of momentum $k$ given by Eqs. (3), (4).

The time integral is performed by contour integration and depends on the position of branch points in the complex plane of $t$. For a supersonic impurity $v^2(k) > v_s^2$ there is one branch point in the upper and one in the lower half-plane, which leads to a double-sided edge singularity

$$\sim \frac{\theta[\omega - \varepsilon(k)] \sin\left(\frac{\Lambda_-^2}{2}\right) + \theta[\varepsilon(k) - \omega] \sin\left(\frac{\Lambda_+^2}{2}\right)}{|\omega - \varepsilon(k)|^\mu},$$

in correlation functions. For a subsonic impurity $v^2(k) < v_s^2$ both branch points are in the upper half-plane, which leads to vanishing of the integral for $\omega < \varepsilon(k)$ and the result

$$\sim \frac{\theta(\omega - \varepsilon(k))\sin\left(\frac{\Lambda_+^2 + \Lambda_-^2}{2}\right)}{|\omega - \varepsilon(k)|^\mu}.$$

In both cases the edge exponent is

$$\mu = 1 - \frac{\Lambda_+^2}{2\pi} - \frac{\Lambda_-^2}{2\pi}. \tag{56}$$

Substituting the relations (54) into this expression one obtains Eq. (5).

## B  Properties of shift functions in Lieb-Liniger model

Similarly to the scattering phase $\theta(\lambda - \mu)$, the shift function obeys $F(\lambda|\mu) = -F(-\lambda|-\mu)$. We can also interchange the arguments using the following non-linear identity

$$F(\lambda|\mu) - F(-\mu|-\lambda) = F(\lambda|q)F(\mu|q) - F(\lambda|-q)F(\mu|-q), \tag{57}$$

obtained by Slavnov in Ref. [52]. In addition, there are useful relation between the density of quasimomenta and the shift function. The first identity

$$1 - 2\pi\rho(\lambda) = F(\lambda|q) - F(\lambda|-q) \tag{58}$$

can be proven by the direct substitution of the left hand side into Eq. (12) and integrating the kernel $K(\lambda - \mu)$ to generate the phase shifts in the right hand side of Eq. (14). Taking $\lambda = q$ one has

$$2\pi\rho(q) = 1 + F(q|-q) - F(q|q). \tag{59}$$

Multiplying both sides by $1 - F(q|q) - F(q|-q)$ and using Slavnov's identity Eq. (57) with $\lambda = \mu = q$ we arrive at another useful relation

$$\frac{1}{2\pi\rho(q)} = 1 - F(q|q) - F(q|-q), \tag{60}$$

which was first established in Ref. [53].

Finally, using Slavnov's identity together with Eqs. (59), (60) and expression (64) for $\rho(q)$ in terms of the Luttinger parameter, we can show that

$$F(q|\lambda) \pm F(-q|\lambda) = -K^{\pm 1/2}\Big(F(\lambda|q) \pm F(\lambda|-q)\Big), \tag{61}$$

which is convenient for interchanging indices in shift functions.

## C  Relation of thermodynamic quantities of Lieb-Liniger model and its Bethe Ansatz solution

It is well known that solutions of Bethe Ansatz equations are related to thermodynamic quantities. These relations are demonstrated in Ref. [33]. For completeness we prove certain thermodynamic relations which we used in Section 4.

We start by taking $\lambda = q$ in Eq. (24) and using Eq. (59) to show that

$$\epsilon'(q)\rho(q) = n, \tag{62}$$

which is identical to Eq. (A.3.6) of Ref. [33].

For small momenta the dispersion relation of elementary excitations becomes linear so one can define the sound velocity

$$v_s = \frac{\partial \epsilon}{\partial k}\bigg|_{k=0} = \frac{\partial_\lambda \epsilon(\lambda)}{\partial_\lambda k(\lambda)}\bigg|_{\lambda=q} = \frac{\epsilon'(q)}{2\pi\rho(q)} = \frac{n}{2\pi\rho^2(q)}, \tag{63}$$

where we used Eqs. (26). This leads to a remarkable exact value of the ground state density of quasimomenta at the edge,

$$2\pi\rho(q) = \sqrt{K}. \tag{64}$$

We obtain another useful relation,

$$\partial_h \epsilon(\lambda) = -2\pi\rho(\lambda), \tag{65}$$

by differentiation of both sides of Eq. (18) with respect to $h$. Consider now the condition $\epsilon(q) = 0$ which determines the dependence $h(q)$. Differentiating both sides of this condition and taking into account Eq. (65) we arrive at

$$\partial_q h = -\frac{\epsilon'(q)}{\partial_h \epsilon(q)} = \frac{\epsilon'(q)}{2\pi\rho(q)} = \frac{n}{2\pi\rho^2(q)} = v_s. \tag{66}$$

This relation can also be obtained from the thermodynamic definition of sound velocity [33]. Eqs. (65), (66) can be used to calculate the derivative of the dispersion relation with respect to $q$. Combining them with Eq. (58) for the density of ground state quasi-momenta we obtain

$$\partial_q \epsilon(\lambda) = \frac{\partial \epsilon}{\partial h}\frac{\partial h}{\partial q} = -2\pi\rho(\lambda)\partial_q h = -v_s\big(1 - F(\lambda|q) + F(\lambda|-q)\big), \tag{67}$$

which is transformed into Eq. (23) with the help of Eq. (61).

# D  Proof of identities in Sections 4.

## D.1  The Resolvent

The equivalence of expressions (15), (16) for momentum and energy of an excitation and the corresponding expressions (17), (18) was proven in Ref. [33]. Below we use a similar method for proving this equivalence for reader's convenience. This will allow us to set up useful notations.

We introduce the *resolvent* $R(\mu, \nu)$ which solves the integral equation

$$R(\nu, \lambda) - \frac{1}{2\pi}\int_{-q}^{q} K(\nu - \mu)R(\mu, \lambda)\,d\mu = \frac{1}{2\pi}K(\nu - \lambda). \tag{68}$$

We rewrite this equation in the operator form

$$\left(\hat{I} - \frac{\hat{K}}{2\pi}\right)\hat{R} = \frac{\hat{K}}{2\pi}, \tag{69}$$

which can be formally inverted leading to

$$\hat{R} = \left(\hat{I} - \frac{\hat{K}}{2\pi}\right)^{-1} \frac{\hat{K}}{2\pi}, \tag{70}$$

or, equivalently

$$\hat{I} + \hat{R} = \left(\hat{I} - \frac{\hat{K}}{2\pi}\right)^{-1}. \tag{71}$$

This identity leads to the formal solution of Eq. (12) for the ground state density of quasi-momenta:

$$2\pi\boldsymbol{\rho} = \left(\hat{I} - \frac{\hat{K}}{2\pi}\right)^{-1} \mathbf{1} = \left(\hat{I} + \hat{R}\right)\mathbf{1}. \tag{72}$$

Here the vector $\boldsymbol{\rho}$ has elements $\rho(\lambda)$ and, similarly, $\mathbf{1}$ has unity elements. The ground state density of rapidities in terms of the resolvent becomes

$$2\pi\rho(\nu) - 1 = \int_{-q}^{q} K(\nu - \mu)\rho(\mu)\,\mathrm{d}\mu = \int_{-q}^{q} R(\nu, \mu)\,\mathrm{d}\mu. \tag{73}$$

Consider now the shift function $F(\nu|\lambda)$ which depends on $\lambda$ as a *parameter*. Taking derivative of both sides of Eq. (14) with respect to $\lambda$ and comparing with Eq. (68) one gets immediately $\partial_\lambda F(\nu|\lambda) = -R(\nu, \lambda)$, so that

$$F(\nu|\lambda) = \pi\rho(\nu) - \int_{-\infty}^{\lambda} R(\nu, \sigma)\,\mathrm{d}\sigma. \tag{74}$$

Here we have used the fact that $F(\nu|-\infty) = \pi\rho(\nu)$. Substituting Eq. (74) into Eqs. (15) gives

$$\pm k = \lambda - 2\pi n + \int_{-\infty}^{\lambda} \mathrm{d}\sigma \int_{-q}^{q} \mathrm{d}\nu R(\nu, \sigma). \tag{75}$$

It is consequence of the theory of linear integral equations that $R(\nu, \sigma) = R(\sigma, \nu)$. Using this fact together with Eq. (72) leads to

$$\pm k = \lambda - 2\pi n + \int_{-\infty}^{\lambda} \mathrm{d}\sigma \int_{-q}^{q} \mathrm{d}\nu K(\sigma - \nu)\rho(\nu)$$
$$= \lambda - \pi n + \int_{-q}^{q} \theta(\lambda - \nu)\rho(\nu)\,\mathrm{d}\nu. \tag{76}$$

Alternatively, the equivalence of Eqs. (15) and (17) can be proven by expressing the scattering phase in the right hand side of Eq. (15) in terms of the shift function using Eq. (14).

We now show the equivalence of (16) and (18). Using Eq. (71) and Eq. (74) we can rewrite the latter as

$$\epsilon(\lambda) - (\lambda^2 - h) = \int_{-q}^{q} R(\lambda, \mu)(\mu^2 - h)\,\mathrm{d}\mu$$
$$= -\int_{-q}^{q} \partial_\mu F(\lambda|\mu)(\mu^2 - h)\,\mathrm{d}\mu$$
$$= 2\int_{-q}^{q} \mu F(\lambda|\mu)\,\mathrm{d}\mu - (q^2 - h)(F(\lambda|q) - F(\lambda|-q)). \tag{77}$$

The integral in r.h.s. is similar to the one in Eq. (16) but has wrong sign and the arguments of the shift function $F$ are interchanged. To manipulate them into the right order we use Slavnov's identity (57). This produces the result (16) plus an extra term,

$$F(\lambda|q)\left(q^2-h-2\int_{-q}^{q}\mu F(\mu|q)\,\mathrm{d}\mu\right)+F(\lambda|-q)\left(q^2-h-2\int_{-q}^{q}\mu F(\mu|-q)\,\mathrm{d}\mu\right). \qquad (78)$$

One recognises $\epsilon(\pm q)$ in the parenthesis of this expression which must be zero by the right choice of chemical potential $h$. We have therefore established the equivalence of Eqs. (16) and (18).

The resolvent operator allows one to prove other useful identities. Let us start with differentiating with respect to $q$ both sides of Eq. (12),

$$\partial_q\rho(\nu)-\frac{1}{2\pi}\int_{-q}^{q}K(\nu-\mu)\partial_q\rho(\mu)\,\mathrm{d}\mu=\frac{1}{2\pi}K(\nu-q)\rho(q)+\frac{1}{2\pi}K(\nu+q)\rho(-q). \qquad (79)$$

Using the fact that $\rho(q)=\rho(-q)$ and Eq. (68) the solution can be found at once:

$$\partial_q\rho(\nu)=\rho(q)(R(\nu,q)+R(\nu,-q)). \qquad (80)$$

Consider now

$$\partial_q n=\partial_q\int_{-q}^{q}\rho(\nu)\,\mathrm{d}\nu=\rho(q)+\rho(-q)+\int_{-q}^{q}\partial_q\rho(\nu)\,\mathrm{d}\nu$$

$$=2\rho(q)+\rho(q)\int_{-q}^{q}(R(\nu,q)+R(\nu,-q))\,\mathrm{d}\nu. \qquad (81)$$

Interchanging the arguments of the resolvent and using Eq. (73) as well as Eq. (64) we get

$$\partial_q n=2\rho(q)+\rho(q)(2\pi\rho(q)-1+2\pi\rho(-q)-1)=4\pi\rho^2(q)=\frac{K}{\pi}. \qquad (82)$$

stated as Eq. (27) in the main text.

## D.2  Derivatives of energy and momentum of excitations in Lieb-Liniger model

By differentiating both sides of Eq. (14) with respect to $q$ and using the expression (69) for the resolvent it is easy to show that

$$\partial_q F(\nu|\lambda)=F(q|\lambda)R(\nu,q)+F(-q|\lambda)R(\nu,-q). \qquad (83)$$

so that

$$\int_{-q}^{q}\partial_q F(\nu|\lambda)\,\mathrm{d}\nu=F(q|\lambda)\int_{-q}^{q}R(\nu,q)\,\mathrm{d}\nu+F(-q|\lambda)\int_{-q}^{q}R(\nu,-q)\,\mathrm{d}\nu$$

$$=(F(q|\lambda)+F(-q|\lambda))(F(q|-q)-F(q|q))$$

$$=(F(q|\lambda)+F(-q|\lambda))(2\pi\rho(q)-1). \qquad (84)$$

Using this identity in Eq. (15) we can show that

$$\pm\partial_q k=-\pi\partial_q n-\int_{-q}^{q}\partial_q F(\nu|\lambda)\,\mathrm{d}\nu-F(q|\lambda)-F(-q|\lambda)$$

$$=-4\pi^2\rho^2(q)-2\pi\rho(q)(F(q|\lambda)+F(-q|\lambda)), \qquad (85)$$

which upon substituting the result (64) becomes Eq. (25).

Consider now the derivative of the dispersion relation with respect to quasimomentum $\lambda$ at fixed $q$. Differentiating both sides of Eq. (18) we get

$$\partial_\lambda \epsilon(\lambda) - \frac{1}{2\pi} \int_{-q}^{q} \partial_\lambda K(\lambda - \mu)\epsilon(\mu) = 2\lambda. \tag{86}$$

Using the property $\partial_\lambda K(\lambda - \mu) = -\partial_\mu K(\lambda - \mu)$ and integrating by parts leads to

$$\partial_\lambda \epsilon(\lambda) - \frac{1}{2\pi} \int_{-q}^{q} K(\lambda - \mu)\partial_\mu \epsilon(\mu) = 2\lambda, \tag{87}$$

where we have used the fact $\epsilon(\pm q) = 0$. The identity (71) allow to write the solution

$$\partial_\lambda \epsilon(\lambda) = 2\lambda + 2 \int_{-q}^{q} \mu R(\lambda, \mu)\, d\mu. \tag{88}$$

Using the fact that $R(\lambda, \mu) = -\partial_\mu F(\lambda|\mu)$ (see Eq. (74)) the integral can be performed by parts leading to

$$\partial_\lambda \epsilon(\lambda) - 2\lambda = -2q\big(F(\lambda|q) + F(\lambda|-q)\big) + 2 \int_{-q}^{q} F(\lambda|\mu)\, d\mu. \tag{89}$$

By virtue of Slavnov's identity, Eq. (57), the integral can be brought to the form which appears in Eq. (15). We obtain

$$\int_{-q}^{q} F(\lambda|\mu)\, d\mu = -\int_{-q}^{q} F(\mu|\lambda)\, d\mu + F(\lambda|q) \int_{-q}^{q} F(\mu|q)\, d\mu - F(\lambda|-q) \int_{-q}^{q} F(\mu|-q)\, d\mu$$

$$= -\lambda + \pi n \pm k(\lambda) + F(\lambda|q)\big(q - \pi n \mp k(q)\big) - F(\lambda|-q)\big(-q - \pi n \mp k(-q)\big).$$

Substituting back into Eq. (89) and using $k(q) = 0$ and $k(-q) = \mp 2\pi n$ gives

$$\partial_\lambda \epsilon(\lambda) = \pm 2k(\lambda) + 2\pi n\big(1 - F(\lambda|q) - F(\lambda|-q)\big). \tag{90}$$

Swapping the indices using Eq. (61) leads to the result (24).

### D.3 Derivatives of energy and momentum of magnon excitations in bosonic Yang-Gaudin model

Differentiating both sides of Eq. (21) with respect to $\lambda$ and integrating by parts gives

$$\partial_\lambda \tilde{\varepsilon} = -\frac{1}{\pi} \int_{-q}^{q} \epsilon(\nu)\partial_\lambda K(2\nu - 2\lambda)d\nu = -\frac{1}{\pi} \int_{-q}^{q} \partial_\nu \epsilon(\nu) K(2\nu - 2\lambda)d\nu. \tag{91}$$

The boundary terms vanish as before since $\epsilon(\pm q) = 0$. Using Eqs. (24), (61) and the fact that $2K(2\nu) = \partial_\nu \theta(2\nu)$ allows us to write

$$\partial_\lambda \tilde{\varepsilon} = -\frac{1}{2\pi} \int_{-q}^{q} \left[ 2k(\nu) + 2\pi n\left(1 + \frac{1}{\sqrt{K}}\big(F(q|\nu) + F(-q|\nu)\big)\right)\right]\partial_\nu \theta(2\nu - 2\lambda)d\nu. \tag{92}$$

Integrating by parts gives

$$\partial_\lambda \tilde{\varepsilon} = -\left(\frac{k(\nu)}{\pi} + n\left(1 + \frac{1}{\sqrt{K}}\bigl(F(q|\nu) + F(-q|\nu)\bigr)\right)\right)\theta(2\nu - 2\lambda)\Big|_{-q}^{q}$$
$$+ \int_{-q}^{q}\left[2\rho(\nu) - \frac{n}{\sqrt{K}}\bigl(R(q,\nu) + R(-q,\nu)\bigr)\right]\theta(2\nu - 2\lambda)\mathrm{d}\nu, \quad (93)$$

where we have used Eq. (26) and the fact that $\partial_\nu F(\mu|\nu) = -R(\mu,\nu)$. The boundary term is calculated using $k(q) = 0$, $k(-q) = -2\pi n$ and the relation

$$F(q|q) + F(-q|q) = -F(q|-q) - F(-q|-q) = 1 - \sqrt{K},$$

which follows from Eqs. (61),(60). It becomes

$$-\left(\frac{k(\nu)}{\pi} + n\left(1 + \frac{1}{\sqrt{K}}\bigl(F(q|\nu) + F(-q|\nu)\bigr)\right)\right)\theta(2\nu - 2\lambda)\Big|_{-q}^{q}$$
$$= -\frac{n}{\sqrt{K}}\bigl[\theta(2q - 2\lambda) + \theta(-2q - 2\lambda)\bigr]. \quad (94)$$

Substituting this result into Eq. (93) and using Eq. (20) leads to

$$\partial_\lambda \tilde{\varepsilon} = 2\tilde{k}(\lambda) - 2\pi n - \frac{n}{\sqrt{K}}\left[\theta(2q - 2\lambda) + \int_{-q}^{q} R(q,\nu)\theta(2\nu - 2\lambda)\mathrm{d}\nu\right]$$
$$- \frac{n}{\sqrt{K}}\left[\theta(-2q - 2\lambda) + \int_{-q}^{q} R(-q,\nu)\theta(2\nu - 2\lambda)\mathrm{d}\nu\right]. \quad (95)$$

The terms in the brackets are just the shift functions

$$2\pi \tilde{F}(\mu|\lambda) = \theta(2\mu - 2\lambda) + \int_{-q}^{q} R(\mu,\nu)\theta(2\nu - 2\lambda)\mathrm{d}\nu, \quad (96)$$

as can be seen by using the resolvent, Eq, (71) to solve Eq. (19). We have therefore demonstrated Eq. (34).

Our next task is Eq. (35). Taking derivative with respect to $q$ of both sides of Eq. (21) and making use of the thermodynamic identities Eqs. (65), (66) gives

$$\partial_q \tilde{\varepsilon} = -\frac{1}{\pi}\int_{-q}^{q} \partial_q \epsilon(\nu) K(2\nu - 2\lambda)\mathrm{d}\nu = \nu_s \int_{-q}^{q} \rho(\nu)\partial_\nu \theta(2\nu - 2\lambda)\mathrm{d}\nu. \quad (97)$$

Integrating by parts results in

$$\partial_q \tilde{\varepsilon} = \nu_s \rho(q)\theta(2q - 2\lambda) - \nu_s \rho(-q)\theta(-2q - 2\lambda) - \nu_s \int_{-q}^{q} \partial_\nu \rho(\nu)\theta(2\nu - 2\lambda)\mathrm{d}\nu. \quad (98)$$

The derivative of the ground state density can be transformed using Eq. (58) into

$$\partial_\nu \rho(\nu) = \frac{1}{2\pi}\partial_\nu\bigl(1 - F(\nu|q) + F(\nu|-q)\bigr) = \frac{\sqrt{K}}{2\pi}\bigl(\partial_\nu F(q|\nu) - \partial_\nu F(-q|\nu)\bigr)$$
$$= -\frac{\sqrt{K}}{2\pi}\bigl(R(q,\nu) - R(-q,\nu)\bigr). \quad (99)$$

Substituting it into Eq. (98) gives

$$\partial_q \tilde{\varepsilon} = \frac{v_s \sqrt{K}}{2\pi} \left[ \theta(2q - 2\lambda) + \int_{-q}^{q} R(q, \nu)\theta(2\nu - 2\lambda)\mathrm{d}\nu \right]$$
$$- \frac{v_s \sqrt{K}}{2\pi} \left[ \theta(-2q - 2\lambda) + \int_{-q}^{q} R(-q, \nu)\theta(2\nu - 2\lambda)\mathrm{d}\nu \right], \quad (100)$$

which is equivalent to Eq. (35) if one uses the representation (96) of the magnon shift function.

The result for the derivative of momentum, Eq. (36) follows immediately from

$$\partial_q \tilde{\varepsilon} = -v_s \partial_\lambda \tilde{k}, \quad (101)$$

which can be shown straightforwardly by inspecting Eq. (97).

Finally, the result in Eq. (37) is obtained by differentiating both sides of Eq. (20)

$$\partial_q \tilde{k} = \pi \partial_q n + \int_{-q}^{q} \partial_q \rho(\nu)\theta(2\nu - 2\lambda)\mathrm{d}\nu + \rho(q)\theta(2q - 2\lambda) + \rho(-q)\theta(-2q - 2\lambda). \quad (102)$$

Using Eqs. (80) and (64) this can be transformed into

$$\partial_q \tilde{k} = K + \frac{\sqrt{K}}{2\pi} \left[ \theta(2q - 2\lambda) + \int_{-q}^{q} R(q, \nu)\theta(2\nu - 2\lambda)\mathrm{d}\nu \right]$$
$$+ \frac{\sqrt{K}}{2\pi} \left[ \theta(-2q - 2\lambda) + \int_{-q}^{q} R(-q, \nu)\theta(2\nu - 2\lambda)\mathrm{d}\nu \right], \quad (103)$$

equivalent to Eq. (37) by virtue of relation (96).

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
