# Peer review of "Mobile impurities in integrable models"

_SciPost Physics, doi:SciPost Phys. 3, 015 (2017)_

## Round 1 · Referee Report · Anonymous (Referee 1) · 2017-1-22

Strengths

  1. The statement 'The equivalence of the edge exponents obtained from the Bethe Ansatz solution and from variation of the dispersion relation was so far established only numerically' is not correct.

Weaknesses

  1. The volume of the manuscript does not correspond to the essentially new content.

Report

The Authors study dynamical correlation functions near a spectral threshold
in one dimensional quantum models. The paper can be divided in three parts.

The first part deals with a phenomenological approach based on the method of mobile impurities. Here the Authors re-derive the edge exponents obtained previously by this method in a series of works.

The most significant part of the manuscript (including appendices) is devoted to integrable systems. Here Lieb-Liniger and Yang-Gaudin models are considered. The elementary excitations in these models are treated as mobile impurities.
This allows the Authors to express the edge exponents in terms of the shift function. The Authors claim that 'The equivalence of the edge exponents obtained from the Bethe Ansatz solution and from variation of the dispersion relation was so far established only numerically'. I do not agree with this statement. I would like to draw the attention of the Authors to the paper [7] where the edge exponents were derived within the Bethe ansatz approach via pure analytical technique. Thus, at least for the Lieb-Liniger model the aforementioned equivalence was already proved, and hence, it cannot be
considered as a new result.

Finally, in the third part of the paper the Authors prove that the phonon backscattering amplitude vanishes for Lieb-Liniger and Yang-Gaudin models. This result confirms a conjecture that in integrable systems this amplitude vanishes due to the existence of infinitely many integrals of motion.

I believe that essentially new results contains only in Section 6 (Phonon backscattering amplitude) and apparently in Eqs. (40) and (49). In all the preceding sections the Authors actually reproduce the results already known. I think that the volume of the manuscript does not correspond to the essentially new content. Therefore, it can be significantly reduced. I also believe that the Authors should provide a more complete description of the results of
[7] and correspondingly adjust some of their statements.

Requested changes

  1. I suggest to reduce the volume of the manuscript. I believe that the equivalence of the edge exponents obtained from the Bethe Ansatz solution and from variation of the dispersion relation was already proved. Therefore, there is no need to prove it again.

  • validity: high
  • significance: ok
  • originality: low
  • clarity: good
  • formatting: excellent
  • grammar: excellent

Author:  Dimitri Gangardt  on 2017-04-26  [id 120]

(in reply to Report 1 on 2017-01-22)

We followed the referee's suggestion and removed the misleading statement
mentioned in his/her report. We have completely rewritten the manuscript,
including its title and abstract. In its new version the manuscript does not
pretend to calculate correlations functions but focuses on mobile impurities
and their interactions with background liquid. Phenomenological parameters
entering theoretical description of the interactions were obtained in Ref. 14,
irrespectively of the edge exponents of correlation functions. Here we only
show that they coincide with chiral phase shifts used to calculate power-law
edge exponents.

The aim of the manuscript is not to calculate edge exponents as in the paper
by Kitanine et al. (Ref. 38) but to express the phenomenological interaction
parameters (collective charges N,J) via Bethe Ansatz solutions of certain
integrable models. Using this result in the expression 8 for the edge
exponents we indeed recover the results of Ref. 38 and we state it explicitly
(references, section and equation numbers are given as they are in the new
version).

We stress, that the goal of our work is to demonstrate the absence of
two-phonon scattering and dissipation. The results of Ref. 38 cannot be used
for this and this necessitates our derivation of expressions 22,33, which
might be lengthy but we cannot think of any shortcuts for the moment. However
we think that the new structure of the manuscript with the discussion of
correlation functions delegated to Appendix is more straightforward and clear.

We hope that the introduction in the rewritten manuscript
reflects this ideas and provides enough motivation for our calculations. In
addition we give full reference to previous results and state, where
necessary, that this is not part of the original research.

---

## Round 1 · Referee Report · Anonymous (Referee 3) · 2017-2-14

Strengths

  • Nice presentation

Weaknesses

  • Claims about novelty are exaggerated

Report

The authors study dynamical correlation functions in one-dimensional
quantum systems. Using a semiclassical analysis, they relate the edge
exponents of power-laws in the response functions to the
phenomenological parameters of their mobile-impurity model. Moreover,
they determine these phenomenological parameters exactly for two
integrable models in terms of the shift functions of their Bethe ansatz
solutions. Finally, they show that the phonon backscattering rate due to
impurities vanishes for integrable models.

The paper is very nicely written. The semiclassical derivation of the
edge exponents in terms of the parameters J and N, and their relation to
the phase shifts obtained from the conventional calculation based on
gauging away the liquid-impurity interaction are well explained and
useful. However, I think the authors need to be more careful about their
claims of novelty. In fact, as far as I know, the predictions of the
mobile-impurity models have been checked analytically for some models.
The claim "The equivalence of the edge exponents obtained from the Bethe
Ansatz solution and from variation of the dispersion relation was so far
established only numerically." is in this generality not correct. But I
think a careful rewording of these claims should suffice to make this
paper acceptable for publication.

Requested changes

  • Make claims about the existence of analytical exponents of Bethe ansatz solvable models more precise.

  • validity: top
  • significance: good
  • originality: ok
  • clarity: high
  • formatting: perfect
  • grammar: perfect

Author:  Dimitri Gangardt  on 2017-04-26  [id 121]

(in reply to Report 3 on 2017-02-14)

We are grateful to the Referee who judges that our paper "is very nicely
written" and its results are "useful". We followed his/her suggestion and made
our claims more precise. In particular, we do not claim to be the first to
obtain edge exponents exactly via Bethe Ansatz solution but rather demonstrate
directly the relation Eq. 22 (new version of the manuscript) between
phenomenological parameter of depleton (mobile impurity model and Bethe Ansatz
Bethe function.

---

## Round 1 · Referee Report · Anonymous (Referee 2) · 2017-2-15

Strengths

The authors prove a relation between solutions of linear integral equations which allows one to make a connection between two apparently different expressions for edge exponents.

Weaknesses

In many cases, the authors present results as if they were discovered by them whereas these are common knowledge. (see report).
The paper is too long regarding to its original content.

Report

The paper "Dynamic correlation functions for integrable mobile impurities" by A.S. Campbell and D.M. Gangardt deals with certain issues related to dynamical correlation functions in the Bose gas and the Yang-Gaudin model.

More precisely, A. Kamenev and L. I. Glazman in "Dynamics of a one-dimensional spinor Bose liquid: A phenomenological approach", predicted a set of universal relations, valid for Galilei invariant models, expressing the edge exponents in terms of partial derivatives of the excitation's dispersion relations.

The work, A. Imambekov and L. I. Glazman "Exact Exponents of Edge Singularities in Dynamic Correlation Functions of 1D Bose Gas" and the work M. B. Zvonarev, V. V. Cheianov and T. Giamarchi "Edge exponent in the dynamic spin structure factor of the Yang-Gaudin model" building on a mixture of non-linear Luttinger liquid theory and Bethe Ansatz expressions for the spectrum argued other kinds of expressions for these edge exponents, this time in terms of the shift function. The work N. Kitanine, K. K. Kozlowski, J. M. Maillet, N. A. Slavnov and V. Terras, "Form factor approach to dynamical correlation functions in critical models" proved, on the basis of exact Bethe Ansatz calculations of correlation functions, the shift function based expressions for the Bose gas.

The present work establishes an equivalence between these apparently different expressions for the edge exponents in the case of the two models.

M. Schecter, D. Gangardt and A. Kamenev in "Dynamics and Bloch oscillations of mobile impurities in one-dimensional quantum liquids" proposed a certain expression for a leading order photon backscattering amplitude that is responsible for the vanishing of a viscous force acting on a moving impurity in a hydrodynamic approximation.

The present work establishes the vanishing of this leading order photon backscattering amplitude in the case of the two models.

The paper stars with a short introduction to the topic which misses the contribution of the integrable model community to the problem: basing on the description of certain citations one may wonder whether the authors did take the time to read the papers they cite (see a more precise discussion below). The introduction also allows the authors to present the problem of interest to the paper.

Section 2 applies the so-called depleton model to compute the exponents in the power-law behaviour of spectral function in terms of quantities they call the depleton charge N and the kink size J. The purpose of this section is not understandable to me. The existence and expression for the edge exponents is well known on the basis of the non-linear luttinger model approach ( Ref[2]) and even on the basis of exact, Bethe Ansatz based calculation (Ref [7]). There is thus low interest in re-deriving these again in terms of some new quantities (N and J). The derivation is rather obscure from the point of view of a novice to the field and useless for an expert. Furthermore, the effective model provided by the authors leads to wrong conclusions in that it does not distinguishes the role played by the velocities of the deep excitations: the conclusions form formula
just above (11) are wrong since the spectral functions may also exhibit two-sided singularities. Its usefullness is thus debatable.

Section 3 establishes a link between the expressions obtained by the authors for the edge exponents in section 2 and those obtained earlier in the literature. Its presence in the paper is only justified by the presence of section 2 which introduces different parameters ($N$ and $J$) than those used earlier in the literature. However, as I stated the presence of section 2 does not seem that useful.

Section 4 presents various Bethe Ansatz issued expressions for the energies and momenta of the excitations in the Bose gas and the Gaudin-Yang model. It leaves one with the wrong impression that some of these expressions where discovered by the authors whereas, in fact, they are known for easily more than 30 years (see below for a deeper explanation).

Section 5 contains the first original result of the paper, namely the proof of relations (12). It is based on establishing certain functional relations satisfied by solutions to linear integral equations. True, some of the identities that are obtained are undoubtedly new. However, this section again leaves one with the wrong impression that most of the expressions where discovered by the authors whereas these have been established a long time ago.

Section 6 applies the results of section 5 to the proof that the so-called phonon backscattering amplitude vanishes for the Bose gas and the Yang-Gaudin model. This is the second original input of this work. However, one should keep in mind that the expression obtained in [40] for this amplitude is only a leading order one. Hence, higher order processes could, in principle, still lead to backscattering/existence of a viscous force. Thus, the conclusions of this section should be softened in that it only provides a first order check.

The paper contains several appendices which contain technical details relative to manipulations of solutions of linear integral equations. Most of the handlings in these appendices are well known. Yet, by the way these are discussed, one gets impression that these were invented by the authors. In fact, most of the proofs presented by the authors can be found in Ref. [3] which is cited at other instance in the paper. It is however not cited as the source for these proofs.

To summarise, the two original results obtained in the paper are interesting and worth publishing. However, the huge "dressing" up of these discoveries by well known content is not necessary, especially that, the way it is written, it produces the wrong impression that this content was also discovered by the authors. I would not object if the paper reproduced known proofs, for the reader's convenience, while making a clear statement that "this is not the original part of the work". However, even then some parts could be shorten. I believe that the original result deserve publication if the content is significantly reduced and the presentation simplified. If these efforts are done, I would recommend the paper for publication.

Below, I list in more details the various problems encountered in this work.

Abstract

-"we recover semiclassically" is a wrong statement since the author's phenomenological approach does not reproduce the existence of two-sided singularities.

  • "relation between these parameters and the Bethe Ansatz shift functions of elementary excitations". This is a miss-leading statement (see discussion in section 3) Thus it should be corrected to something of the sort " We establish a rigorous relation between Galilei invariance based predictions for the edge exponents and exact Bethe Ansatz calculations.''Which is closer to the truth.

  • "implies the absence of phonon back scattering". This is an overstatement in that it is only proved to leading order.

Introduction

  • "Progress in this direction has been recently reported in Refs.[7, 8, 9, 10, 11]." The statement is wrong and shows the lack of knowledge of the content of the cited works. Regarding to the progress in the calculation of correlation functions and matrix elements of local operators, one could cite the important works of the Kyoto group on elementary blocks or further developments on qKZ equations and fermionic bases, the resolution of the inverse problem by the Lyon group or the calculation of thermal correlators by the Wüppertal group to name a few. The cited works [7, 8, 9, 10, 11] characterise, on the basis of exact calculations, various critical behaviours of the correlation functions in the XXZ spin chain or the Bose gas. In particular, the work [7] characterises various dynamical properties of the correlators in the Bose gas -the edge singular behaviour of spectral functions in particular.

  • Again "Another approach was undertaken in Refs. [12, 13, 14] where the matrix elements where calculated numerically." is an huge understatement of the content of these works. Not only the matrix elements where computed there but, in fact, a full numeric characterisation of the spectral functions was achieved, in particular by showing the critical excitation tresholds.

  • Reference [15] is correct but I trust that the original credit should be given to Luther and Peschel and Haldane.

  • "This description is sufficient to calculate reliably the static correlation functions such as the one-body density matrix and the corresponding momentum distribution [17]." The use of reliably for an approximate method sounds strange to me. In any case, this was achieved on the basis of exact rigorous calculations much earlier: see the works of the Kyoto group and of Tracy-Vadiya.

  • "It is known that the LL theory fails to describe correctly the dynamical correlations even in the low energy limit [2]." The reference to [2] is very strange. This is known from much much earlier works on dynamical correlators in the XX chain.

-- paragraph 2 on page 3. The authors do not cite ref [7] where it was shown that the edge exponents can be computed exactly for integrable models.

Section 2

  • "It is well known (see Ref.[2]" parenthesis missing.

  • The arguments of the section are unclear and look like an ad-hoc procedure to get the result. The conclusions of the method are not correct since one misses the existence of two-sided singularities (check [2]).

  • The sentence "In the next section we show that this result is in complete agreement with the results of Refs. [18, 30, 46]." omits the contributions of the integrable model community.

Section 3

  • The authors introduce the parameters $N$ and $J$ in their depleton model. These parametrised the edge exponents as computed within their method. By requiring consistence of their depleton model's conclusion with the predictions for the edge exponents issuing from the reasonings based on the non-linear Luttinger liquid model and arguments of Galilean invariance, they identify J and N with partial derivatives of the excitation's dispersion relation. Then, after making this identification, they claim to recover the various predictions, in terms of the phase shifts, that appeared in the literature.
    However, from the very start, their parameters where tuned in a way to correspond to the phase shift predictions in the first place. Thus, this section does not really contain any derivation.

  • For all these reasons, I believe that it would be reasonable to start the results of the paper from this section and simply state the expressions for the pĥase shifts (12) as predicted by Galilean invariance. And then to discuss the exact results obtained in [7], and earlier predicted on the basis of a mixture of Bethe Ansatz calculation and non-linear Luttinger liquid theory in [30], in what concerns the Bose gas, and predictions in [33] by similar means for the Gaudin-Yang model.

Section 4

-The denomination "shift function" for the solution to (22) is unusual, this object is traditionally called "dressed phase" or scattering phase, see ref [3] equation (4.39). The matter is that F used by the authors only identifies with the shift functions in very specific cases, as actually discussed in ref [3].

  • All integral representations present in this section are well known. They are all listed and proven in, say ref [3]. For instance (23)-(24) is equation (4.6)-(4.7) in [3], while (25) is equation (4.28) and (4.19) in [3], (26) is (4.9). In particular the equivalence between these representations is proven in [3]. For this reason, the statement "The proof of equivalence of (25), (26) and (23), (24) can be found in Appendix C.1" is very missleading. I repeat that I have no problem if the authors recall the proofs (although these are of not much use to the "original" part of the research presented in the paper) but they have to very clearly state they they only recall the proof of a known result.

  • I have no immediate reference in mind for the similar types of representations in the Gaudin-Yang model, but these are also very well known results (and the proof is similar). The authors should thus give the relevant credit.

Section 5

-"The results of previous section allows us to obtain the dispersion relation ε(k) of excitation in Lieb-Liniger and Yang-Gaudin models in terms of shift functions F (ν|λ) and F̃ (ν|λ) correspondingly". The sentence is missleading since it insinuates that there were some new results in section 4 whereas it is just a reminder of well known facts.

  • "Indeed, the previous studies [30, 33] suggested" this disregards that relation (30) was established through exact calculations for the Bose gas.

  • Most relations proposed in section 5.1 and 5.2 are well known. What makes the original part is only their combination to check the nice prediction (12). However, the authors should not give themselves the credit of having established some specific differential relations.

  • In the paragraph below (35) the authors reintroduce a mass $m$, eventhough they stated taking $m=1/2$ at some earlier point. They should make up their mind once for all.

  • The parenthesis in (46), (47), (48) are too large.

Section 6

  • One should keep in mind that the results of [40] stem form a linearisation procedure, hence an approximation of some more complex equation. As such, $\Gamma_{+-}$ only represents the leading contributions to the viscous force. Higher order scattering contributions may also contribute to the effect. Hence, showing that $\Gamma_{+-}=0$ only shows that to the leading order such effects do not take place. It is still an interesting, but weaker, result.

Section 7

  • "respect to the Bethe Ansatz parameters mentioned above ." too much space

  • "We expect that due to the general structure of Bethe Ansatz equations our results can be generalised to other models soluble by nested BA, such as , the fermionic Hubbard model and integrable spin chains." I disagree with the conclusions. The authors use a trick that heavily depends on the bare energy being quadratic, which fails for more complex models.

Appendices

  • Most of the relations established in the appendices have already been established and a major part thereof is even extremely well known. The authors give proofs without making it explicit in the text that it is only a reminder of known facts is missleading.

  • Some non-exhaustive examples:

  • The authors cite [50] to credit it for (55). Then they use it to prove (58). Yet (58) was proven in 1998 in the paper of Korepin and Slavonv "The New Identity for the Scattering Matrix of Exactly Solvable Models".

  • Above (74), the fact that the resolvent kernel is symmetric is not proven in [50]. It is a simple consequence of the theory of linear integral operators.

  • (75) the proof that can be found in [3].

  • The authors should thus make a clear identification of what is new and what is only a copying of known results.

Requested changes

see report

  • validity: ok
  • significance: ok
  • originality: ok
  • clarity: poor
  • formatting: below threshold
  • grammar: excellent

Author:  Dimitri Gangardt  on 2017-04-26  [id 122]

(in reply to Report 4 on 2017-02-15)

We are thankful to the Referee for reading carefully our manuscript and providing us with such a detailed guidance for its improvement. Before we turn to the specific questions raised by the referee we would like to stress that the manuscript has been completely re-organised and its main parts were substantially rewritten. The title and the abstract were modified accordingly and do not anymore suggest that we pretend to calculate known dynamical correlation functions.

Here are the main points:

  1. The Referee is right by presenting the first result of our manuscript as a proof of equivalence between different approaches to edge exponents in one-dimensional quantum liquids. We state it explicitly in the introduction of the revised version.

  2. We agree that the inelastic processes are considered to the leading, two-phonon, order. We now state it explicitly throughout the text.

  3. We took into account the Referee's unsatisfaction with Section 2 and followed his advice to start the manuscript with Section 3. Section 2 was revised and its content became Appendix A as it is indeed not crucial for the main message of the manuscript. We feel, however, that our approach based on path integral provides a new insight on the mobile impurity model and connects to our previous work in which dynamics of mobile impurities was obtained using this technique. Our method shows that the power-law singularities are nothing but a semiclassical approximation to the path integral justified by the logarithmically large action of phonons and we recover all known results this way.

In the first version of the manuscript it was implicitly assumed that the velocity of the depleton in Section 2 (Appendix A in the new version) is smaller than sound velocity leading to one-sided singularity only. The missing two-side singularities alluded to by the Referee do appear in our approach under a careful analysis of the time integration contour which we provide in the revised version.

  1. The collective charges $N$, $J$ are not "tuned in a way to correspond to the phase shift predictions" as Referee claims. They were introduced in Ref.[14] (in the new version of the manuscript) from first principles as thermodynamic response to changes in background density and current. Of course their equivalence to the chiral phase shifts and the corresponding edge exponents is not coincidental and the reason for this is explained by semiclassical calculations in Appendix A (former Section 2). We have rewritten Section 3 to clarify this point.

We now address the Referee's remarks in more detail. References, section and equation numbers are given as they are in the new version unless stated otherwise.

Abstract

The abstract was completely rewritten and the misleading statements
    were removed.

Introduction

Following the critique of the Referee we have changed the scope of the
manuscript which is now concentrated on the interaction parameters of
mobile impurities with the phononic background, so called collective
charges. The reference to previous results on dynamical correlations
are only given in this context. We are grateful to Referee for
pointing them out but feel that discussion of some of these
works will be outside the scope of the manuscript.

We give credits  Luther and Peschel as well as Haldane for Luttinger
Liuquid.

We do not discuss static correlation functions in the new version of
the manuscript

We describe results of the work by Kitanine et al. (Ref. 38) in
context of our findings.

Appendix A (Section 2 in the old version)

The typo was corrected.

The method we use is essentially a semiclassical approximation for
path integral following method pioneered by Iordanskii and Pitaevskii
(Ref. 50) and it is justified by the logarithmically large action in the
vicinity of the excitation threshold. The value of the collective
charges are not chosen ad hoc to get the desired result, but rather
obtained following procedure outlined in Ref. 14.  The resulting edge
exponents are indeed identical to those obtained earlier, this is
explained in Section 2 (new version). The unfortunate missing of
two-sided singularities in the previous version is now rectified and
properly explained.

The sentence "In the next section ..." does not exist in the new
version.

Section 2 (3 in the old version)

We addressed collective charges in main point 4 above.

We followed the Referee advice to state the relations of the chiral
phase shifts to the dispersion relation. We do it after introducing
the depleton model from Ref. 14 and the definition of collective
charges obtained there. We explain that  comparing the two expressions
leads to Eq. 7. We postpone the discussion of the exact results of
Ref. 38 (former 7) and their relations with the results of mobile
impurity models (Refs. 22, 37) until the BA shift functions
are introduced in Sec. 3

Section 3 (4 in the old version)

We used the denomination "shift function" following Refs. 22 and
37. We state the alternative denomination " dressed phase" in the
introduction to avoid confusion.

To stress that all results in this section were obtained previously we
added the sentence "Below we present main equations which allow us to
obtain dispersion relation $\varepsilon(k)$ used for calculation of
the collective charges. All results of this Section can be found in
Refs. [32, 45] and are reproduced here to make presentation
self-contained." We cite books by Korepin et al. and by M. Gaudin.

Section 4 (5 in the old version)

The first sentence of this section is changed to avoid
misinterpretation of the results in the previous section as new.

In our manuscript the chiral phase shifts (and their combinations N,J)
are defined by Eqs. 6 (3,4). Their expressions in terms of shift
functions, Eq. 22  did not appear in literature to
the best of our knowledge. Ref. 38 which contains exact calculations
of power-law asymptotics of correlation functions does indeed provide
edge exponents directly in terms of shift functions, however it does
not relate them to the chiral phase shifts. The goal of Sec. 4 (new
version) is to establish Eq. 22 directly. We have added a sentence
explaining this point and the relation of our results to those of
Ref. 38.

Here the referee does not substantiate his claim that "most relations
proposed ... are well known". We couldn't find them in the literature
like Ref. 32 and had to derive them ourselves. If the
referee knows of any publications containing these relations we will
be happy to accommodate the corresponding references. Also, we do not
claim the differential relations, which are rather technical,
to be results of our work.

We reintroduce the mass m to be able to compare the results of Bethe
Ansatz calculations, which are usually done with m=1/2, with Eq. 6. It
is a standard practice to reintroduce physical units in the end of
calculation.

The size of parenthesis was decided  automatically by LaTeX and we
rather leave it to this judgement.

Section 5 (6 in the new version)

We have changed wording in this section to stress that back-scattering
vanishes in the leading, two-phonon order.

Section 6 (7 in the old version)

"too much space": it was probably LaTeX glitch, so in the new version 
the space seems to be reasonable.

Ref. 17 deals with models without Galilean invariance and obtains
dissipation rates by using very similar  methods to ours. The only
difference is that dependence of energy on background velocity needed
to derive expressions 3,4 for collective charges is not automatically
provided by Galilean invariance. However this dependence can still be
calculated via Bethe Ansatz and incorporated in our formalism.

Appendices

Appendix A is an appropriately  shortened version of Sec. 2 of the
old manuscript. It also contains a more careful analysis of
two-sided power-law singularity.

Appendix B

We cite the '98 paper by Korepin and Slavnov just after Eq. 60
(former 58) to give credit to this earlier work.

We are grateful to referee for his remark on the symmetry of the
kernel and have incorporated it in the text.

Appendix C

We state explicitly that derivation of
the expressions in this Appendix can be found in Ref. 32.

Appendices D.1 We give full credit to Ref. 32 for the prove of equivalence of different expressions for momentum and energy of excitations.

Appendices D.2 and D.3

As we said above, the differential relations for energy and
momentum are new to the best of our knowledge.

---

## Round 2 · Referee Report · Anonymous · 2017-4-27

Strengths

1. The paper contains new important results.

Weaknesses

1. I do not see evident weaknesses.

Report

I appreciate the changes made by the authors and recommend the manuscript for publication.

Requested changes

1. No changes required.

---

## Round 2 · Referee Report · Anonymous · 2017-5-23

Strengths

Nicely written paper containing important results.

Weaknesses

No obvious weaknesses.

Report

The authors have responded adequately to the questions raised by the referees in the first round. I recommend the paper for publication.

Requested changes

No changes required.

---

## Round 2 · Referee Report · Anonymous · 2017-5-25

Strengths

-

Weaknesses

-

Report

In their work, Campbell and Gangardt re-derive some important results of the theory of Nonlinear Luttinger liquids. Most notably, they re-affirm the absence of viscosity for excitations describing the edge of the spectral continuum in two specific integrable models: the Lieb-Liniger model and the Yang-Gaudin model for BOSONS with spin 1/2.

While there is little new physics in their work, I believe it is methodically useful and should be published in SciPost. The previous Referees already fought heavily for assigning the proper credit to various groups, so I spare the Authors from some minor grievances.

Requested changes

I have two specific requests to the Authors:

(1) In the Introduction (p. 3), please substantiate of modify the phrase: "The equivalence
of these results to those obtained in Ref. [37] using phenomenological mobile impurity model
relies on the conjectured relation between the collective charges and BA shift functions which
we demonstrate here." It is not clear how the notion of "collective charges" was involved in demonstrating the equivalence.

(2) It seems to me that the Authors consider only the bosonic Yang-Gaudin model. If this is the case, that must be clearly stated in the beginning of the manuscript and repeated in the main text in the sections devoted to that model.

I hope the Authors introduce changes adequately addressing these two comments.

  • validity: high
  • significance: good
  • originality: good
  • clarity: high
  • formatting: excellent
  • grammar: excellent

Author:  Dimitri Gangardt  on 2017-06-10  [id 141]

(in reply to Report 3 on 2017-05-25)

Referee 149 made two requests:

(1) In the Introduction (p. 3), please substantiate of modify the phrase:
"The equivalence of these results to those obtained in Ref. [37] using
phenomenological mobile impurity model relies on the conjectured relation
between the collective charges and BA shift functions which we demonstrate
here." It is not clear how the notion of "collective charges" was involved in
demonstrating the equivalence.

Our response:

We have removed this phrase and slightly modified the preceding sentence.
See also response to

(2) It seems to me that the Authors consider only the bosonic Yang-Gaudin
model. If this is the case, that must be clearly stated in the beginning of
the manuscript and repeated in the main text in the sections devoted to
that model.

Our response:

The referee is absolutely right and we have replaced Yang-Gaudin model by
bosonic Yang-Gaudin model throughout the manuscript.

---

## Round 2 · Referee Report · Anonymous · 2017-5-31

Strengths

As in the first report

Weaknesses

Some citations are strange.

Report

The authors did implement many of the referees' suggestions what has improved the paper. However, they also did made some quite queer amendments, especially in what concerns the bibliography and the citations.

In the present version, there is no mention -with the exception of reference 37- of all the efforts that where made within the integrable model community regarding to the calculation of correlation functions and extractions of the edge exponents, at least in some regimes.
The authors already did provide some citations in the first version of the paper and I have suggested some other literature, so as to provided a fairer picture. Since, in fine, the author's work relies directly on integrable techniques, I believe that making such an omission is unfair.

The authors inaccurately describe the content of certain papers. To be more precise:

In the middle of page 3, the authors discuss the goal of their paper. There they claim that their result "reproduces the conjectured identity of chiral linear combinations of the collective charges, so called chiral phase shifts, with BA shift functions".
This is not true. Paper 37 expresses the edge exponents in terms of shift functions by arguing directly that phase shifts in the unitary transform should be given by the same formula as for non-interacting fermions, and this is enough for their calculation.
The works 22,36 identify the coupling constants in their effective Hamiltonian by comparing their low-energy spectrum with the one issuing from the Bethe Ansatz. By the way, the authors do not prove the predictions issuing, per se, from the work 36 since it deals with the XXZ chain which is not considered in the author's paper.

I stress that, all these works directly provide an expression for the edge exponents in terms of the shift function. Thus, 37 and 38 provide the same expression for the edge exponents.

What the authors do is to prove the expressions for the phase shifts that were argued to hold in ref. 23 and in "Phenomenology of One-Dimensional Quantum Liquids Beyond the Low-Energy Limit" by A. Imambekov and L.I.Glazman. Curiously, that last paper is not cited in the present version of the paper while it was cited in the first one.

Page 4 The authors attribute to 14 the proposal of the relations (3)-(4). However that paper dates to 2012 while "Phenomenology of One-Dimensional Quantum Liquids Beyond the Low-Energy Limit" appeared in 2009, just as reference 23. Again, true, the expressions obtained there are written in another language but I believe that the correspondence between the two is not that hard.

Below (60). The identity was conjectured to hold much much earlier than [52]. However, the work [52] was the first to establish it.
See, e.g. "Conformal dimensions in Bethe ansatz solvable models" in what concerns the nested Bethe ansatz case.

Requested changes

see report

  • validity: ok
  • significance: ok
  • originality: ok
  • clarity: ok
  • formatting: acceptable
  • grammar: excellent

Author:  Dimitri Gangardt  on 2017-06-10  [id 142]

(in reply to Report 4 on 2017-05-31)

Response to Referee 153

(1) In the present version, there is no mention -with the exception of
reference 37- of all the efforts that where made within the integrable model
community regarding to the calculation of correlation functions and
extractions of the edge exponents, at least in some regimes. The authors
already did provide some citations in the first version of the paper and I
have suggested some other literature, so as to provided a fairer
picture. Since, in fine, the author's work relies directly on integrable
techniques, I believe that making such an omission is unfair.

Our response

In the newer version of the manuscript we tried to get the main message across
and skip unnecessary discussion irrelevant to our main results: the relation
of phenomenological parameters N, J and Bethe Ansatz shift fuctions. This was
done following Referees' suggestion to streamline the
discussion. Unfortunately, this resulted in vanishing of many citations from
the exactly integrable community not directly relevant for this manuscript.

(2) In the middle of page 3, the authors discuss the goal of their
paper. There they claim that their result "reproduces the conjectured identity
of chiral linear combinations of the collective charges, so called chiral
phase shifts, with BA shift functions". This is not true. Paper 37 expresses
the edge exponents in terms of shift functions by arguing directly that phase
shifts in the unitary transform should be given by the same formula as for
non-interacting fermions, and this is enough for their calculation. The works
22,36 identify the coupling constants in their effective Hamiltonian by
comparing their low-energy spectrum with the one issuing from the Bethe
Ansatz.

Our response

The argument of paper by Imambekov and Glazman assumes implicitely that
parameters of the effective mobile impurity are given in terms of BA shift
functions. It is indeed done by evoking free fermions and their effective
phase shifts, but no mathematical proof is given. Papers by Cheianov and
Pustilnik as well as Zvonarev, Cheianov, Giamarchi do not provide proof
either. The 2012 review [1] states that the equivalence between chiral phase
shifts given in terms of the impurity dispersion and BA shift functions
(dressed phases) is an open problem. Referee is right by saying that the form
(6) of the chiral phase shifts first appeared in in the publication
"Phenomenology of One-Dimensional Quantum Liquids Beyond the Low-Energy Limit"
by A. Imambekov and L.I.Glazman, but as it is explained in this paper
parameters of the effective mobile impurity model used in all the cited
references can be obtained from the impurity dispersion. Here we demonstrate
the missing relation of the parameters (in the form of N, J) of effective
impurities to the BA shift functions. In a sense this is nothing but the
"comparing their low-energy spectrum with the one issuing from the Bethe
Ansatz".

We are grateful to Referee for pointing out the omission of the above paper by
Imambekov and Glazman, which is now Ref. 32 in the new version of the
manuscript.

(4) Page 4 The authors attribute to 14 the proposal of the relations
(3)-(4). However that paper dates to 2012 while "Phenomenology of
One-Dimensional Quantum Liquids Beyond the Low-Energy Limit" appeared in 2009,
just as reference 23. Again, true, the expressions obtained there are written
in another language but I believe that the correspondence between the two is
not that hard.

Our response

By no means we claim that Ref [14] (co-authored by one of us)
was the first to establish relations between effective impurity parameters
and its dispersion. Indeed, the relations (3)-(4) are equivalent to Eq. (6)
under linear transformation (7) from N,J to the chiral phases as we point out
in our work citing the appropriate references. However, the depleton model,
(1), (2) was formulated in Ref. [14] in terms of collective charges N,J using
a different approach and we prefer to start with (3)-(4) rather than
deriving them from (6) and (7).

(5) Below (60). The identity was conjectured to hold much much earlier than
[52]. However, the work [52] was the first to establish it.

Our response

We are grateful to Referee for pointing it out and have modified the sentence
around Eq. (60) correspondingly.

---

## Round 2 · Author Response

We have changed the overall scope of the paper, which is now focused on
calculations of effective interactions parameters (collective charges) of a
mobile impurity (depleton) model. We mention their relation to the edge
exponents in various correlation functions, but Section 2 in the previous
version, containing extended derivation of the correlation functions'
asymptotics, is moved now to Appendix A.

We now clearly state which results were obtained previously and give
appropriate credit. We still feel, however, that for the sake of completeness
and reader's convenience, these results should be included in the manuscript
and we state this in the text.

---

## Round 2 · List of Changes

* Title and abstract changed

* Introduction rewritten

* Section 2 moved to Appendix A

* Credit to previously obtained results is given throughout the paper.

---

## Round 3 · Referee Report · Anonymous (Referee 1) · 2017-6-27

Strengths

No stregths

Weaknesses

No weakness

Report

I have no remarks to the new version of the manuscript and recommend it for publication.

Requested changes

No requested changes

---

## Round 3 · Referee Report · Anonymous (Referee 3) · 2017-6-28

Strengths

see previous report

Weaknesses

No obvious weaknesses

Report

The paper can be published in its present form.

Requested changes

-

---

## Round 3 · Referee Report · Anonymous (Referee 2) · 2017-7-9

Strengths

see previous report

Weaknesses

no new comments

Report

The paper can be published in its present form.

Requested changes

No changes.

---

## Round 3 · Author Response

Dear editor,

We would like to resubmit our manuscript after some minor revisions.
The revised version is now available at http://arxiv.org/abs/1701.00810v3.

We hope that the revised manuscript is now suitable for publication in
SciPost.

Yours sincerely,

Andrew S. Campbell
Dimitri M. Gangardt

---

## Round 3 · List of Changes

• The phrase "The equivalence of these results to those obtained in ..." is removed and the preceding sentence is modified.

  • Yang-Gaudin model is replaced by "bosonic Yang-Gaudin model" everywhere in the text.

  • Reference [38] added.

  • We have modified the sentence around Eq. (60).

---

## Editorial Decision

published